# Hyperbaric Oxygen Treatment—From Mechanisms to Cognitive Improvement

**DOI:** 10.3390/biom11101520

**Published:** 2021-10-15

**Authors:** Irit Gottfried, Nofar Schottlender, Uri Ashery

**Affiliations:** 1School of Neurobiology, Biochemistry and Biophysics, Life Sciences Faculty, Tel Aviv University, Tel Aviv 6997801, Israel; iritgo@tauex.tau.ac.il (I.G.); schottlender@mail.tau.ac.il (N.S.); 2Sagol School of Neuroscience, Tel Aviv University, Tel Aviv 6997801, Israel

**Keywords:** hyperbaric oxygen treatment (HBOT), cognition, brain disorders, neuroprotection, neuroinflammation, Alzheimer’s disease

## Abstract

Hyperbaric oxygen treatment (HBOT)—the medical use of oxygen at environmental pressure greater than one atmosphere absolute—is a very effective therapy for several approved clinical situations, such as carbon monoxide intoxication, incurable diabetes or radiation-injury wounds, and smoke inhalation. In recent years, it has also been used to improve cognition, neuro-wellness, and quality of life following brain trauma and stroke. This opens new avenues for the elderly, including the treatment of neurological and neurodegenerative diseases and improvement of cognition and brain metabolism in cases of mild cognitive impairment. Alongside its integration into clinics, basic research studies have elucidated HBOT’s mechanisms of action and its effects on cellular processes, transcription factors, mitochondrial function, oxidative stress, and inflammation. Therefore, HBOT is becoming a major player in 21st century research and clinical treatments. The following review will discuss the basic mechanisms of HBOT, and its effects on cellular processes, cognition, and brain disorders.

## 1. Hyperbaric Oxygen Treatment (HBOT): The Concept

HBOT—the medical administration of 100% oxygen at environmental pressure greater than one atmosphere absolute (ATA)—is used clinically for a wide range of medical conditions. One of HBOT’s main mechanisms of action is elevation of the partial pressure of oxygen in the blood and tissues as compared to simple oxygen supplementation [1,2]. This allows five to ten times more oxygen to enter the blood plasma and to reach tissues suffering from low oxygen supply (following, e.g., brain injury, stroke, or vascular dysfunction). Therefore, it is not surprising that HBOT has been used for over 50 years for wounds (non-healing diabetic foot ulcers), air embolisms or decompression sickness, burned tissue repair, carbon monoxide intoxication, peripheral arterial occlusive disease, smoke inhalation, radiation injury, and promoting recovery from serious illness [3,4,5,6,7,8,9,10]. Nevertheless, today, there are only 13 FDA-approved HBOTs [11]; however, in parallel, there are a growing number of “off-label” uses, which have not been cleared by the FDA, such as treatment for stroke patients or patients suffering from Alzheimer’s disease (AD) [12,13], and even treatment of COVID-19 patients, which have shown very promising results [14,15,16,17,18,19]. Further clinical trials that are currently in progress, and additional basic scientific studies aimed at understanding HBOT’s mechanisms of action, will most probably expand the use of HBOT to other areas.

## 2. Cognitive Improvement

### 2.1. Cognitive Improvement Following Brain Injuries

Although the use of HBOT in cases of brain-related disorders is pending FDA approval, there are numerous studies showing improved cognitive assessment following treatment for several brain injuries [20]. For example, post-stroke patients suffer from reduced cognitive performance, and in particular, memory difficulties. HBOT for stroke patients at late chronic stages has shown significant improvements in all memory measures. These clinical improvements are well-correlated with improvements in brain metabolism, mainly in temporal areas. High oxygen (92%) alone was also shown to positively affect the working memory of individuals with intellectual and developmental disabilities, at least in the short-term [21]. Similar improvement was seen in a large cohort of post-stroke patients who underwent 40 HBOT sessions (2 ATA), leading to significant neurological and cognitive improvements, even at the late chronic phase after stroke [22,23]. Mechanistically, in preclinical studies, HBOT has been suggested to reduce oxidative stress, inflammation, and neural apoptosis, thereby improving functional recovery from stroke [24]. It was also suggested that HBOT in rats suffering from ischemic stroke stimulates the expression of trophic factor and neurogenesis, and the mobilization of bone marrow stem cells to the ischemic area, which can enhance cell repair [25]. In addition, HBOT elevates cerebral blood flow (CBF), associated with restoration of physical abilities and cognitive functions [26,27]. The improvement in cognition and executive functions, as well as in physical abilities, gait, sleep, and quality of life in these stroke patients continued for up to three months after the last treatment, which was the follow-up period in that study [27]. These encouraging results suggest the occurrence of long-term changes, lasting the order of months. Similarly, in patients with mild traumatic brain injury (TBI), HBOT improved hippocampal CBF [28] and facilitated recovery during the rehabilitation phase [29]. Moreover, growing evidence suggests that HBOT can induce neuroplasticity and improve cognitive function in patients suffering from chronic neurocognitive impairment due to TBI, stroke, and anoxic brain damage [22,23,30,31,32]. These changes were associated with the induction of cerebral angiogenesis, increased CBF and volume, and improved cerebral white and gray microstructures [33]. 

Other teams have investigated whether HBOT can improve brain function and cognition in neurodegenerative diseases such as AD and vascular dementia (VD), and if HBOT can also affect healthy people or improve cognitive decline in the elderly who are suffering from cognitive impairments.

### 2.2. Cognitive Improvement Following HBOT in AD and VD

Recent human studies have shown that HBOT can improve cognitive functions in patients with mild cognitive impairment (MCI), AD, and VD [13,20,34,35,36,37,38], and ameliorate the reduced brain metabolism in MCI and AD [34,35]. Similarly, cerebrovascular disease patients showed improvement in motor and cognitive performance compared to a control group following HBOT [38]. Interestingly, improvements in cognitive function assessed by Mini-Mental State Exam (MMSE) and Mini-Cog test were reported in AD patients even one month after the end of the last HBOT, and for up to three months in amnestic MCI patients. In addition, HBOT ameliorated the reduced brain glucose metabolism in some of the AD and amnestic MCI patients [34]. These are very promising results, because they suggest that even with severe cognitive deterioration in progressive neurodegenerative brain disorders, relatively short-duration HBOT (40 min once a day for 20 days) can improve conditions for one to three months. In a more severe case of AD, a longer treatment of eight weeks (1.15 ATA) reversed the patient’s symptomatic decline and PET scan showed an increase in brain metabolism [35]. Nevertheless, the current belief is that HBOT cannot revert severe cases with major neuron loss and therefore should be considered mainly at early disease stages, when only minimal cognitive deficiency is detected. It should be noted that the elevation of pressure by itself was also suggested to regulate AD [39]. However, further research in this direction should explore the exact effect. A larger group of VD patients who received 12 weeks of HBOT (2 ATA) showed improvement in MMSE scores and elevated serum humanin levels [36]. Humanin is a unique human mitochondrion-derived peptide that has neuroprotective effects [40,41,42] and, together with findings of increased brain metabolism, this suggests an important role for improving mitochondrial function as part of HBOT’s mechanism of action. As HBOT use in the clinic is considered to be safe and well-tolerated, it should be considered and recommended as an alternative therapeutic approach for AD and VD [37], as well as in early stages of MCI. Hence, HBOT improves several aspects of brain activity including an improvement in cerebral blood flow, brain metabolism, and brain microstructure, and this leads to improvement in cognitive functions and physical functions, sleep, and gait leading to an overall improved performance (Figure 1). Nevertheless, it is also clear that although the effects of HBOT last, in some studies, for several months, when treating patients with progressive neurodegenerative diseases such as AD, maintenance HBO treatments will probably be needed.

### 2.3. Cognitive Improvement in Healthy Individuals

Over the last few decades, several studies have examined the possible contribution of HBOT to cognitive performance in both young and elderly populations. In one of the first studies examining the effects of HBOT on the elderly [43], it was found to improve cognitive function in elderly patients with cognitive deficits. In a more recent study with a cohort of healthy young adults, HBOT increased spatial working memory and memory quotient, and this was correlated with changes in regional homogeneity as measured by resting-state functional MRI [44]. In another prospective study, double-blind randomized healthy volunteers were asked to perform a cognitive task, a motor task and a simultaneous cognitive–motor task (multitasking) while in a functional HBO chamber. Compared to the performance under normobaric conditions, single cognitive and motor task, and multitasking performance scores were significantly enhanced by the HBO environment, supporting the hypothesis that oxygen is a rate-limiting factor for brain activity [45]. These results were further validated by two recent studies that examined the effects of HBOT on healthy young [46] and old [47] adults. In these studies, HBOT resulted in an improved learning curve and higher resilience to interference of episodic memory in the healthy young adults [46], and induced cognitive enhancements in healthy aging adults, which were associated with regional improvement in CBF [47]. Similarly, in a recent paper, a group of elderly patients with memory loss at baseline to HBOT showed improved cognitive performances following 60 daily HBOT sessions (2 ATA) and this was associated with an increase in CBF [48]. Interestingly, when HBOT was applied for a short time (only 15 consecutive days), there was no improvement in cognitive impairment in the elderly [49], suggesting that a longer treatment is necessary. Indeed, current protocols are extending the treatment to two to three months (40–60 daily sessions, 5 days per week, 2–3 ATA) and promise to yield more significant and long-lasting effects [12]. 

In summary, it is clear that the HBO environment, in and of itself, improves cognitive performance, and that this can be attributed directly to the elevated oxygen levels, suggesting that oxygen is a rate-limiting factor for brain activity [45]. However, repeated exposure to HBOT for longer periods of time is needed to achieve long-lasting effects that lead to changes in vascular, neuronal, and cellular activity, as detailed in Figure 2 [12].

## 3. Mechanistic Explanation for the Effects of HBOT on Cognition

What are the cellular and molecular pathways that contribute to the long-term neuron, function- and cognition-enhancing effects of HBOT? A series of studies using animal models for brain injuries and brain diseases showed an improvement in the animals’ cognitive performance and provided a mechanistic understanding of some of HBOT’s effects. Not surprisingly, these effects are not mediated by a single pathway, but were found to be mediated by several pathways, including inhibition of apoptosis, improvement of mitochondrial function, stem cell proliferation, enhancement of antioxidant defense activity, reduction in neuroinflammation, and neuroprotection (Figure 2). The “normobaric oxygen paradox” or “hyperoxic–hypoxic paradox” has been suggested to play a key role in HBOT’s effects [12,50,51,52]. It is based on the fact that during HBOT sessions, oxygen level is increased from 21 to 100% (or less in some cases) and at the end of each treatment, oxygen level is reduced back to 21%. Such fluctuations activate several factors: elevation of oxygen can activate nuclear factor erythroid 2-related factor 2 (Nrf-2), while the reduction to 21% can be interpreted as a hypoxic signal and activate Hypoxia-Inducible Factor 1-alpha (HIF1a) [50,51]. HIF1a belongs to a family of proteins that are involved in angiogenesis and vascular remodeling, erythropoiesis, glycolysis, iron transport, and survival [53,54,55]. Nrf2 is involved in several cellular defense mechanisms, and it mediates the repair and degradation of damaged proteins [51,55,56], and activates the antioxidant pathways and the detoxification of endogenous and exogenous products [57]. Under high hyperoxia, nuclear factor kappa B (NF-κB), which is usually activated under oxidative stress and inflammation, is also activated [51], and mediates inflammatory and immune responses. NF-κB is also involved in synaptic plasticity and in the antiapoptotic pathway by activating Bcl-2 [58]. Some of these effects are discussed below. It should be noted that the optimal conditions for achieving best results from the “Hyperoxic–Hypoxic Paradox” require additional research in the coming years.

### 3.1. Mitochondrial Function

Mitochondria consume roughly 85 to 90% of the oxygen that we breathe and are the major source of ATP production. It is therefore likely that the main molecular target of HBOT is the mitochondrion. As already noted, humanin, a neuroprotective mitochondrion-derived peptide in humans, was elevated in VD patients following HBOT [36], suggesting a major role for mitochondrial activity in HBOT’s mechanisms of action. Recent studies have suggested the therapy’s direct effects on neurons were mediated by mitochondrial transfer from cell to cell. HBOT was shown to facilitate the transfer of mitochondria from astrocytes to neuronal cells, making the latter more resilient to neuroinflammation [59]. This neuroglial crosstalk may facilitate recovery and explain some of the mechanisms induced by HBOT [50]. In TBI rats, HBOT for 4 h (1.5 ATA) led to an increase in ATP levels and neuron survival, both of which were associated with improved cognitive recovery [60]. Furthermore, in a rat model for AD, HBOT reduced mitochondria-mediated apoptosis signaling by increasing Bcl-2, which is anti-apoptotic, and decreasing Bcl-2-associated X protein (Bax), which is pro-apoptotic [61].

### 3.2. Neurogenesis and Angiogenesis 

An additional avenue for cognitive improvement might be stem cell proliferation. Stem cell proliferation has been documented on various occasions following HBOT [62,63,64], and evidence for neuronal cell proliferation has emerged in the last two decades. In an early study, HBOT for hypoxic ischemic neonatal rats promoted neurogenesis of endogenous neuronal stem cells, as measured by an increase in both 5-bromo-2′-deoxyuridine (BrdU) and doublecortin, in the subventricular zone (SVZ) and the hippocampal dentate gyrus (DG)—an area involved in spatial navigation [65]. Accordingly, HBOT improved spatial learning and memory abilities in rats with TBI [66]. This was associated with an increase in hippocampal neuronal activity. 

These results were further supported by another study in which HBOT induced neuronal cell proliferation, as revealed by an increase in nestin and BrdU in the hippocampal DG area [67] and elevation of Wnt-3 and nestin in the SVZ [68]. In a study aimed at examining the mechanistic contribution of HBOT to recovery from TBI, it was found that HBOT increases neuronal stem cell proliferation and migration to the lesion area, as well as the levels of vascular endothelial growth factor (VEGF) and its receptor VEGFR-2, Raf-1, Mitogen-activated protein kinase (MEK1/2), and phospho-extracellular signal-regulated kinase (ERK) 1/2 protein [69]. Accordingly, it was suggested that HBOT promotes neuronal stem cell proliferation and possibly angiogenesis through VEGF/ERK signaling [69]. Moreover, in a rat model for VD, HBOT also stimulated neurogenesis in the piriform cortex and improved blood supply [70]. HBOT was also shown to enhance mobilization of bone marrow stem cells to an ischemic area and the release of trophic factors that can promote brain and neuronal recovery and enhance neurogenesis [25]. Interestingly, in patients with delayed encephalopathy after acute carbon monoxide poisoning, HBOT mobilized, circulating stem cells in the peripheral blood, which was associated with improved cognition [71]. In a TBI rat model, HBOT stimulated angiogenesis as evidenced by a higher number of BrdU- and VEGF-positive cells, and an increase in the number of BrdU- and NeuN-positive cells, suggesting enhanced neurogenesis [72]. These findings provide support for improvement of human brain cognition associated with changes in cerebral angiogenesis and neuronal growth and proliferation improving CBF and brain activity [33]. Indeed, a recent study showed that HBOT improves blood flow in an AD mouse model by mitigating the blood vessel constriction that occurs in these AD mice under the regular course of the disease but without HBOT. This was associated with an improved performance of the AD mice [48]. Moreover, in elderly patients with significant memory loss at baseline, HBOT increased CBF and improved cognitive performance [48]. It would be interesting to examine whether HBOT also restores neurogenesis in neurodegenerative diseases such as AD, and whether it will affect neurogenesis and angiogenesis [73] in wild-type mice and healthy humans.

### 3.3. Neuroinflammation 

Another important effect of HBOT in several brain dysfunctions is reduced neuroinflammation. TBI is usually associated with increased inflammation, apoptosis and gliosis, neuronal cell death, and cognitive and motor dysfunction. In a TBI rat model, HBOT was shown to reduce neuroinflammation and increase levels of the anti-inflammatory cytokine interleukin (IL)-10; these changes were associated with improvements in cognitive deficit [72]. In an AD mouse model, HBOT reversed hypoxia and ameliorated brain pathology, and improved the animals’ behavioral performance [74,75]. This improvement was also associated with a reduction in proinflammatory cytokines such as IL-1b, IL-6, and tumor necrosis factor alpha (TNFα), and an increase in anti-inflammatory cytokines such as IL-4 and IL-10, leading to reduced neuroinflammation. HBOT also significantly improved recovery from sepsis following cecal ligation and puncture; the treatment was associated with a reduction in the inflammatory response, including decreased expression of TNFα, IL-6, and IL-10 [17,76]. Changes in cytokines following exposure to oxygen have also been reported in humans. A low-intensity exercise program in combination with exposure to mild hyperoxia (30%) elevates the proinflammatory IL-6 that contributes to host defense during infection and tissue, while at both mild (30% oxygen) and high hyperoxic state (100% oxygen), the anti-inflammatory cytokine IL-10 was elevated significantly [52]. In a rat model for MCI, HBOT had a protective effect on early cognitive dysfunction that was mediated by ERK. These animals performed better in the Morris water maze, and showed less apoptosis and better hippocampal cell morphology [77]. In a rat model for AD that was induced by injections of amyloid β peptide into the hippocampus, HBOT improved animal behavior, and reduced neuronal damage, astrocyte activation, and dendritic spine loss. This was associated with a reduction in hippocampal p38 mitogen-activated protein kinase (MAPK) phosphorylation [78], which occurs in the early stage of the disease and is associated with increased neuroinflammation, cytoskeletal remodeling, and tau phosphorylation [79,80]. These papers suggest that the MAPK/ERK pathways, which are involved in cell proliferation and plasticity, are also a target for HBOT. 

### 3.4. Neuroprotective, Antioxidant, and Antiapoptotic Activities

HBO preconditioning induced tolerance to cerebral ischemia [81]. This was mediated by an increase in SIRT1, a class III histone deacetylase, which has been suggested to be involved in neuroprotection [82]. The neuroprotective effect of preconditioning HBOT was associated with a reduction in lactate dehydrogenase and was attenuated by a reduction in SIRT1 activity or expression by either the SIRT1 inhibitor EX527 or SIRT1 knockdown. Interestingly, the neuroprotective effect was mimicked by resveratrol, a SIRT1 activator. Changes in SIRT1 level were also associated with elevation in B-cell lymphoma 2 (Bcl-2) expression and a decrease in cleaved caspase 3 level, suggesting that some of the effects might be mediated via inhibition of apoptosis [82]. Moreover, expression of SIRT1 in the brain was associated with increased expression of the nuclear factor erythroid 2-related factor 2 (Nrf-2), heme oxygenase 1 (HO-1), and superoxide dismutase 1 (SOD1), whereas the level of malondialdehyde (MDA) decreased, supporting the notion that HBOT enhances the antioxidant defense pathway, thereby assisting in neuroprotection [83]. Indeed, HBO preconditioning increased the expression of SIRT1, Nrf-2, and HO-1 and ameliorated memory dysfunction in additional models of cognitive decline [84], and SIRT1 was also shown to play a role in recovery after middle cerebral artery occlusion in rats. Therefore, this might serve as the mechanism for HBOT’s effects in cases of acute ischemic stroke [85]. A combination of HBOT and Ginkgo biloba extract following induction of toxicity with amyloid β (Aβ fragments) demonstrated enhanced SOD and glutathione levels, while levels of MDA and Bax, and activity of caspases 9 and 3 were reduced in rat hippocampal tissue, suggesting both antioxidant and antiapoptotic activity [61,86]. In a mouse model for mild TBI, HBOT improved learning abilities and prevented astrocyte activation and neuronal loss, suggesting a neuroprotective effect [87]. Additional involvement in apoptotic pathways was demonstrated in an AD rat model that showed improved cognitive and memory abilities following HBOT, which were associated with NF-κB pathway activation and reduced hippocampal neuron loss [88]. Further animal model studies may reveal additional mechanisms underlying the effects of HBOT, thus facilitating the development of more efficient HBOT protocols. Taken together, HBOT has a multifaceted neuroprotective effect on the brain that involves the immune, neuronal and vascular systems, leading to enhancement and recovery of cognitive performance.

## 4. HBOT—The Next Leap

HBOT has been used for centuries to treat a variety of symptoms and syndromes, and in recent years, it has been shown to improve many brain disorders. Nevertheless, it is still not fully established clinically, and additional basic research and clinical trials are necessary. Notably, in recent years, numerous such clinical trials have been supported by the NIH. Over 230 clinical trials examining HBOT have been reported (https://clinicaltrials.gov/, accessed on 4 September 2021). Of these, 50 clinical trials are examining the effects of HBOT on brain-related injuries and disorders. Current and future clinical trials will provide additional validated information for a wider range of disorders, while basic research will expand our mechanistic understanding and help optimize treatment conditions by allowing for more accurate determinations of treatment length, frequency of treatments, and the exact protocol. This will reduce cost, time, and complications. Overall, HBOT is becoming a central player in the 21st century healthcare system with the ability to improve both personal performance and cognition.

## Figures and Tables

**Figure 1 biomolecules-11-01520-f001:**
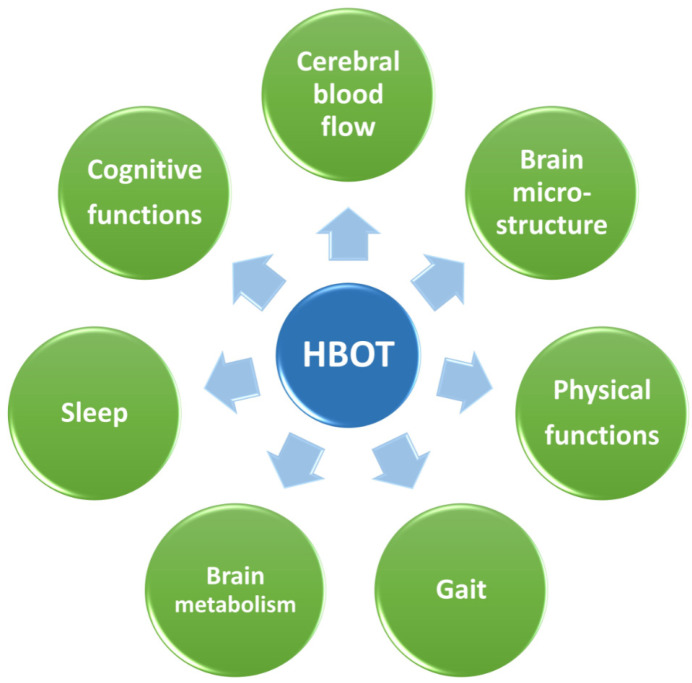
HBOT improves brain function. HBOT has been shown to improve cerebral blood flow, brain metabolism, and brain microstructure, leading to improved cognitive functions, physical functions, sleep, and gait.

**Figure 2 biomolecules-11-01520-f002:**
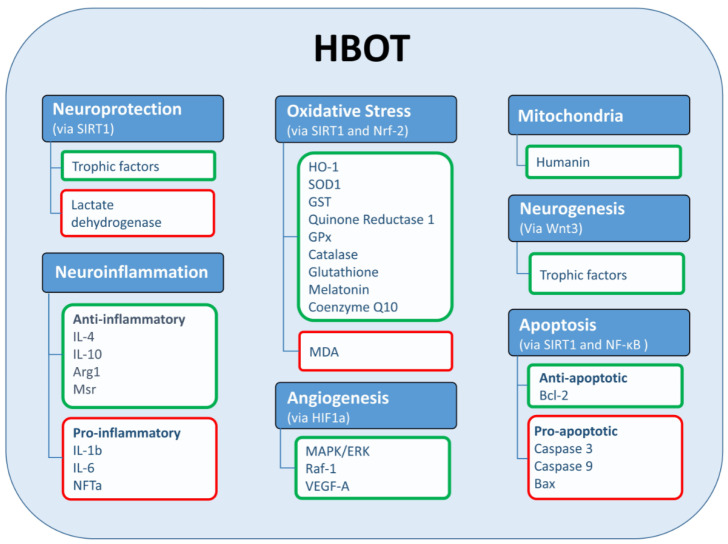
HBOT affects multiple cellular and molecular pathways. HBOT affects several molecular and cellular pathways that are important for cellular and neuronal recovery including neuroprotection via SIRT1, oxidative stress via SIRT1 and Nrf-2, apoptosis via SIRT1, neurogenesis via Wnt3. Green frames represent proteins and processes that are upregulated; red frames represent proteins and processes that are downregulated. Abbreviations: nuclear factor erythroid 2-related factor 2 (Nrf-2), nuclear factor kappa B (NF-κB), Hypoxia-Inducible Factor 1-alpha (HIF1a), heme oxygenase 1 (HO-1), superoxide dismutase 1 (SOD1), malondialdehyde (MDA), B-cell lymphoma 2 (Bcl2), Bcl-2-associated X protein (Bax), vascular endothelial growth factor (VEGF-A), Glutathione-S-transferases (GST), Glutathione Peroxidase (GPx), tumor necrosis factor alpha (TNFa), Wnt Family Member 3 (Wnt3).

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
