# Peer review of "Hyperbaric Oxygen Treatment—From Mechanisms to Cognitive Improvement"

_biomolecules, 2021, doi:10.3390/biom11101520_

Round 1

Reviewer 1 Report

The topic of this manuscript falls within the scope of Biomolecules Journal.  It is quite interesting paper. The following review will discuss the basic mechanisms of HBOT, and its effects on cellular processes, cognition, and brain disorders. The appropriate figures have been provided. The article is easy to read and logically structured.   

There are only few comments in the reviewer opinion which should be taken under consideration by the authors:

  1. In the introduction the Authors should presented more preciosuly in which diseases HBO may be applied:
  2. a) Wound healing [Thermal imaging and planimetry evaluation of the results of chronic wounds treatment with hyperbaric oxygen therapy. Adv Clin Exp Med 2019, 28 (2), 229-236; Thermal Effects of Topical Hyperbaric Oxygen Therapy in Hard-to-Heal Wounds – A Pilot Study, J. Environ. Res. Public Health2021, 18(13), 6737
  3. diabetic foot [Gębala-Prajsnar K., et al.Selected physical medicine interventions in the treatment of diabetic foot syndrome. Acta angiologica, 2015, 21,4,140-145;.
  4. peripheral arterial occlusive disease [Hyperbaric Oxygen Therapy Enhanced Circulating Levels of Endothelial Progenitor Cells and Angiogenesis Biomarkers, Blood Flow, in Ischemic Areas in Patients with Peripheral Arterial Occlusive Disease. J Clin Med. 2018;7(12):548.]
  5. decompression sickness (Ascent rate, age, maximal oxygen uptake, adiposity and circulating venous bubbles after diving. J Appl Physiol 2002, 93 1349–56.

Author Response

We thank the referee for the constructive comments and suggestions.

Comment #1

In the introduction the Authors should presented more preciosuly in which diseases HBO may be applied.

We have updated the list of treatments and their references according to the referee's suggestions. We also replaced the FDA approved treatments reference with a link to the FDA.gov site.

Reviewer 2 Report

Just update references to most recent peer-review publications.

Author Response

We thank the referee for the constructive comments and suggestions.

Comment #1

Just update references to most recent peer-review publications.

We have now included additional and recent peer-review publications as the referee has suggested (Ref # 3-10, 39, 50-58,73).

Reviewer 3 Report

I really appreciate this sum up manuscript on hyperbaric oxygen therapy.

I have just some minor remarks:

Line 259 -> In the middle of the line some weird signs are present (at least on my copy)

The authors are several times mentioning hyperbaric oxygen therapy in different settings and treatments, it would be of interest for the reader to specify the pressure applied and not only the number of sessions, since they are not always identical.

On other parts of the manuscript references are made to NF-KB and NRF2, some interesting considerations on their interplay and oxygen levels can be found in the following reference and will add to the mechanisms explained. Some other consideration on inflammatory responses as well:

Fratantonio D, Virgili F, Zucchi A, Lambrechts K, Latronico T, Lafère P, Germonpré P & Balestra C. (2021). Increasing Oxygen Partial Pressures Induce a Distinct Transcriptional Response in Human PBMC: A Pilot Study on the "Normobaric Oxygen Paradox". Int J Mol Sci 22.

Bosco G, Paganini M, Giacon TA, Oppio A, Vezzoli A, Dellanoce C, Moro T, Paoli A, Zanotti F, Zavan B, Balestra C & Mrakic-Sposta S. (2021). Oxidative Stress and Inflammation, MicroRNA, and Hemoglobin Variations after Administration of Oxygen at Different Pressures and Concentrations: A Randomized Trial. International Journal of Environmental Research and Public Health 18, 9755.

Balestra C, Lambrechts K, Mrakic-Sposta S, Vezzoli A, Levenez M, Germonpré P, Virgili F, Bosco G & Lafère P. (2021). Hypoxic and Hyperoxic Breathing as a Complement to Low-Intensity Physical Exercise Programs: A Proof-of-Principle Study. International Journal of Molecular Sciences 22, 9600.

On the Alzheimer’s diseases side, it is interesting to mention that two different approaches can be applied with hyperbaric oxygen therapy one more on oxygen levels (mentioned by the authors) and one, more amazing, on the pressure applied....here are 2 references to help.

Lanzillotta C, Di Domenico F, Perluigi M & Butterfield DA. (2019). Targeting Mitochondria in Alzheimer Disease: Rationale and Perspectives. CNS Drugs 33, 957-969.

Denis PA. (2013). Alzheimer's disease: a gas model. The NADPH oxidase-Nitric Oxide system as an antibubble biomachinery. Med Hypotheses 81, 976-987.

The authors have presented a nice work helping the readers to better understand the benefits of oxygen therapy, one useful addition would be to discuss the “frequency” of oxygen sessions in some sentences, perhaps in the next leap session. See previous  refs.

I would like to thank the author for such an educational work, well deserved in the field.

Author Response

We thank the referee for the positive feedback and the constructive comments and suggestions.

Comment #1

Line 259 -> In the middle of the line some weird signs are present (at least on my copy)

Corrected.

Comment #2

The authors are several times mentioning hyperbaric oxygen therapy in different settings and treatments, it would be of interest for the reader to specify the pressure applied and not only the number of sessions, since they are not always identical.

We have specified the pressure applied in those sections that we have also elaborated on the number of HBOT session assisting the readers to have a more comprehensive picture of the treatment protocol.

Comment #3

On other parts of the manuscript references are made to NF-KB and NRF2, some interesting considerations on their interplay and oxygen levels can be found in the following reference and will add to the mechanisms explained. Some other consideration on inflammatory responses as well:

We have included a new section discussing the effects on NRF2, NF-KB and HIF1a in the “Mechanistic Explanation” paragraph and cited the publications the referee mentioned.

Comment #4

On the Alzheimer’s diseases side, it is interesting to mention that two different approaches can be applied with hyperbaric oxygen therapy one more on oxygen levels (mentioned by the authors) and one, more amazing, on the pressure applied....here are 2 references to help.

We have mentioned the possibility, indeed amazing, to affect AD only by changing the pressure applied. 

Comment #5

The authors have presented a nice work helping the readers to better understand the benefits of oxygen therapy, one useful addition would be to discuss the “frequency” of oxygen sessions in some sentences, perhaps in the next leap session. See previous  refs.

We have touched this point in the “Next Leap” section. Additionally, this review will be published back to back with another review we are currently writing that covers aspects of mitochondrial function and oxidative stress in HBOT. In this review we are discussing more lengthily the concept of intermittent HBOT.